# Identification of a New Insulin-like Growth Factor 3 (*igf3*) in Turbot (*Scophthalmus maximus*): Comparison and Expression Analysis of IGF System Genes during Gonadal Development

**Chunyan Zhao** [1,2,†], **Sujie Zheng** [1,†], **Yongji Dang** [1], **Mengshu Wang** [1] **and Yichao Ren** [1,*]

1   School of Marine Science and Engineering, Qingdao Agricultural University, Qingdao 266109, China
2   Department of Biological Sciences, National University of Singapore, Singapore 117543, Singapore
*   Correspondence: yichaor@126.com or yichaoren@qau.edu.cn
†   These authors contributed equally to this work.

**Abstract:** Fish reproduction is closely related to the regulation of the brain and liver, making it essential to identify the factors that control this process. The turbot (*Scophthalmus maximus*) is an economically significant species that has been successfully breeding through industrial aquaculture. Investigation of factors into the involvement of gonadal development is crucial for artificial breeding. In this study, a new insulin-like growth factor 3 *igf3* gene was cloned and characterized. Additionally, all three types of turbot IGFs contain a distinct IGF domain, with IGF3 and IGF2 being grouped with other teleosts, demonstrating a closely related genetic relationship. The expression analysis showed that *igf3* mRNA is predominantly expressed in the gonad and brain (specifically in the pituitary and hypothalamus), suggesting its effects at multiple levels in the brain–pituitary–gonadal axis. Furthermore, the mRNA levels of *igfs* during gonadal development were examined. In the gonad and liver of female turbots, the expression levels of *igfs* mRNA significantly increased from stage II to VI during the process of oogenesis, including maturation and degeneration. In the testis and liver of male turbots, *igf3* mRNA maintained high expression levels during the proliferation of spermatogonia at stages II and III. In addition, the highest levels of *igf1* and *igf2* were observed at the beginning of spermatogenesis and during sperm production at stages V and VI. All the results suggest that the IGFs were closely related to the gonadal development in turbot and improve a better understanding of the IGF system in the regulation of gonadal development in teleost.

**Keywords:** *Scophthalmus maximus*; insulin-like growth factor 3; brain; gonadal development

**Key Contribution:** This study characterizes a new *igf3* in turbot, analyzes the differential expression in gonad and liver during gonadal development in breeding season, and contributes to a better understanding of the potential roles of IGF system in turbot breeding and reproduction.

## 1. Introduction

Reproduction is an essential life process for animals, ensuring the preservation and continuity of genetic information within populations. The brain–pituitary–gonad (BPG) axis plays a vital role in regulating reproduction [1], involving feedback regulation of gonadal hormones on target organs in teleost fish. Previous research has demonstrated that the reproductive system demands substantial amounts of energy-related biomolecules, such as vitellogenin and lipids, which are primarily derived from the liver [2]. Consequently, reproductive success is closely linked to growth and energy metabolism [3].

The liver has been reported to regulate fish gonad development by modulating the BPG axis [2,4,5]. Various growth factors, including insulin-like growth factor (IGF), influence gamete production, and energy metabolism during gonad development [1,4]. Research has shown that IGF is a highly conserved polypeptide throughout the evolutionary process and plays a crucial role in fish reproduction [6,7]. Typically, IGF1 and IGF2 have been identified

in a wide variety of teleost, and similarly to mammals, are produced in the liver [7]. The IGF1 has been shown to directly impact the follicle-stimulating hormone (FSH) [8]. Additionally, it is widely accepted that IGF1 and IGF2 can affect gonadal development and germ cell proliferation in male and female gonads through endocrine function when released into circulation [7,9].

Previous studies have shown that a novel *igf3* has been identified in zebrafish (*Danio rerio*), exhibiting unique expression patterns in the gonad and brain. The *igf3* levels progressively increase in zebrafish follicle cells [10,11] and exhibit strong regulatory effects on gonadotropin analogs and human chorionic gonadotropin (HCG) [9]. Two transcripts of *igf3* have also been discovered in zebrafish [11]. Studies in tilapia (*Oreochromis niloticus*) [12,13] and orange-spotted grouper (*Epinephelus coioides*) [13,14] have also revealed that *igf3* mRNA is present in the gonads and brain during the early stages of sex determination and differentiation. Moreover, IGF3 can work synergistically with FSH to stimulate spermatogonial proliferation and differentiation [14]. This evidence suggests that IGF3 is involved in gonadal and reproductive development in teleost.

Turbot (*Scophthalmus maximus*) possesses significant economic value in both Europe and China. Previous studies have indicated that turbot reaches sexual maturity at two years of age [15]. Although *igf1* and *igf2* have been isolated in turbot, and their expression patterns were examined in the regulation of ovarian development [8], a comprehensive characterization of the turbot IGF system remains elusive. In this study, a new *igf3* in turbot was identified and characterized. Additionally, we analyzed cross-species variations in IGFs compared to other teleost. Concurrently, the expression levels of *igfs* in the gonad and liver were investigated during the gonadal development of turbot. These findings could contribute to a better understanding of the potential roles of the IGF system in breeding and reproduction in turbot.

## 2. Materials and Methods

### 2.1. Fish Sampling

All turbots used in the study were obtained from a fish farm in Laiyang, China. Turbot is raised under suitable conditions with temperature of $17 \pm 0.5$ °C, dissolved oxygen of $7.55 \pm 0.05$ mg/L, and fed with fresh miscellaneous fish twice a day. During the stages of gonadal development, 25 female turbots and 25 male turbots were anesthetized and sampled. Half of each gonad was placed in Bouin's solution for 24 h until the histology analysis. The remaining half was frozen in liquid nitrogen for qPCR for the expression analysis. The body weight, liver weight, and gonad weight were measured during the experiment. The gonadal somatic index (GSI) defined as the ratio of gonad weight to body weight and liver somatic index (LSI) defined as the ratio of liver weight to body weight were calculated. All experimental procedures were conducted strictly in accordance with the research protocols approved by Qingdao Agricultural University for the ethical treatment of experimental animals.

### 2.2. Histological Observation

The fixed gonadal tissues were dehydrated by a series of gradient alcohol, transparented by xylene, and embedded by paraffin. Then the paraffin blocks were cut into 5 μm sections. After being stained with hematoxylin and eosin, the slices were observed with a microscope (Axioscope 5, Zeiss, Oberkochen, Germany).

### 2.3. igf3 Isolation

The testis tissue at stage II was prepared, and the total RNA from testis was extracted, using SPARK easy Improved Tissue/Cell RNA Kit (SparkJade, Qingdao, China). The purity of RNA extracts was detected with UV spectroscopy. First-strand cDNA was synthesized followed by a transcript first-strand cDNA synthesis kit (TranStart, Zaozhuang, China). A conserved fragment of *igf3* based on the gene-specific primers was obtained with the PCR amplification procedures using $2 \times$ Taq Plus Master Mix II (Dye Plus) (Vazyme, Nanjing,

China). Then, a series of RACE-PCRs according to a SMARTer RACE 5′/3′ Kit (Takara, Tokyo, Japan) were performed to obtain the full length of *igf3*. The purified PCR products were sequenced with corresponding inserts into the pMD18-T cloning vector (Takara, Tokyo, Japan). All the primers used are listed in Table 1.

**Table 1.** Primers used for gene cloning and expression analysis.

| Aim | Primer | Primer Sequence (5′–3′) | Length |
|---|---|---|---|
| Partial cloning | *igf3*-1F | CTGTGCCAAACCAAAGAGCC | 504 bp |
| | *igf3*-1R | CAAACTGCTGTGCCGTGTC | |
| | *igf3*-2F | CGGAGTGTGAGGAGTGTGTG | 411 bp |
| | *igf3*-2R | GGTGGTTTTCTGTGGGCTCT | |
| Cloning 5′-end (RACE) | *igf3*-GSP5-1 | AGGAGGAATGGTGGTTTTCTGTG | |
| | *igf3*-GSP5-nested | GGTGGTTTTCTGTGGGCTCT | |
| Cloning 3′-end (RACE) | *igf3*-GSP3-1 | GAGAAATACTGTGCCAAACCAAAGAG | |
| | *igf3*-GSP3-nested | CACAGAAAACCACCATTCCTCCT | |
| expression analysis | *igf1*-F | CTGCTGAGGTTAAAGTGCGA | 165 bp |
| | *igf1*-R | AAGCCTCTCTCTCCACACAC | |
| | *igf2*-F | GAGACGCTGTGCGGAGGAGA | 196 bp |
| | *igf2*-R | TTTCAGACTTGGCGGGTTT | |
| | *igf3*-3F | GTACGGATCTCCTCAGCGAC | 180 bp |
| | *igf3*-3R | GGCTCTTTGGTTTGGCACAG | |
| | *β-actin*-F | ATCGTGGGGCGCCCCAGGCACC | 543 bp |
| | *β-actin*-R | CTCCTTAATGTCACGCACGATTTC | |

*2.4. Phylogenetic and Bioinformatics Analyses*

The amino acid sequence of turbot IGF3 was deduced using BioEdit (Ibis Biosciences, Carlsbad, CA, USA). The protein sequences of IGF1 and IGF2 of turbot and IGFs in other teleost were collected from National Center of Biotechnology Information website (http://www.ncbi.nlm.nih.gov/, accessed on 15 July 2022). The multi-aligned turbot IGFs (IGF1, IGF2, and IGF3) were performed by BioEdit (Ibis Biosciences, Carlsbad, CA, USA). The domain was predicted by SMART (https://smart.embl-heidelberg.de/, accessed on 15 July 2022), and the three-dimensional structure was predicted by RCSB PDB (https://www.pdbus.org/, accessed on 20 July 2022). The phylogenetic analyses were conducted by MEGA7 software (version 7.0, Mega Limited, Auckland, New Zealand) using neighbor (NJ)-based molecular evolution method and was visualized in iTOL(https://itol.embl.de/itol_account.cgi, accessed on 20 July 2022). The position and orientation of *igfs* and their adjacent genes in humans, chickens, zebrafish, and medaka (*Oryzias latipes*) on the chromosome were determined using NCBI database for syntenic analysis.

*2.5. Quantitative Real-Time PCR (qPCR)*

The qPCR was performed using 2 × SYBR Green qPCR Mix (SparkJade, Shandong, China) a quantitative thermal cycler (Quantagene, Beijing, China). Specific primers of *igfs* and reference genes of *β-actin* are listed in Table 1. The qPCR mixture reaction is a total of 10 μL, with 5 μL of 2 × SYBR qPCR Mix, 3.4 μL of ddH$_2$O, 0.4 μL of forward primers (10 μM), and 0.4 μL of reverse primers (10 μM), and 0.8 μL of cDNA template. The procedure of PCR reaction was performed with 3 min at 94 °C;10 s at 94 °C and 30 s at 58 °C for 40 cycles; 10 s at 95 °C, 5 s at 65 °C. The relative gene expression levels were analyzed using the method of $2^{-\Delta\Delta CT}$.

### 2.6. Statistical Analysis

Statistical analysis was performed using SPSS version 26.0, and all the results showed as mean $\pm$ SEM. Data were analyzed using two-way analysis of variance (ANOVA) followed by Duncan's multiple range tests. Differences were considered to be significant at $p < 0.05$.

## 3. Results

### 3.1. Gonadal Histology and Reproductive Stages Assessment

Based on the histology, five gonadal stages of II to VI were identified during the breeding season of turbot (Figure 1A–J). In females, the oocyte development of five stages were primary growth oocytes and perinucleolar oocytes at stage II (Figure 1A), beginning the early vitellogenesis with increasing lipid granules at stage III (Figure 1B), undergoing late vitellogenesis with lipid and yolk granules filling most of the ooplasm at stage IV (Figure 1C), ovulated oocytes with completed vitellogenesis at stage V (Figure 1D), and residual postovulatory follicle in the degenerated ovaries at stage VI (Figure 1E), respectively. In males, the spermiogenesis of five stages was observed and assessed by the major type of male germ cell (Figure 1F–J). At stage II, the testis was filled with spermatogonia with the largest diameters (Figure 1F). Following stage III, the spermatogonia developed into primary spermatocytes which had a smaller diameter but a relatively bigger nucleus (Figure 1G). At stage IV, the testis was filled with secondary spermatocytes with deep staining, and a litter number of haploid spermatids appeared (Figure 1H). At stage V, spermatozoa with smaller sizes and darker staining were in the seminiferous tubules (Figure 1I). Then at stage VI, the testis was empty with spermatozoa expelled out.

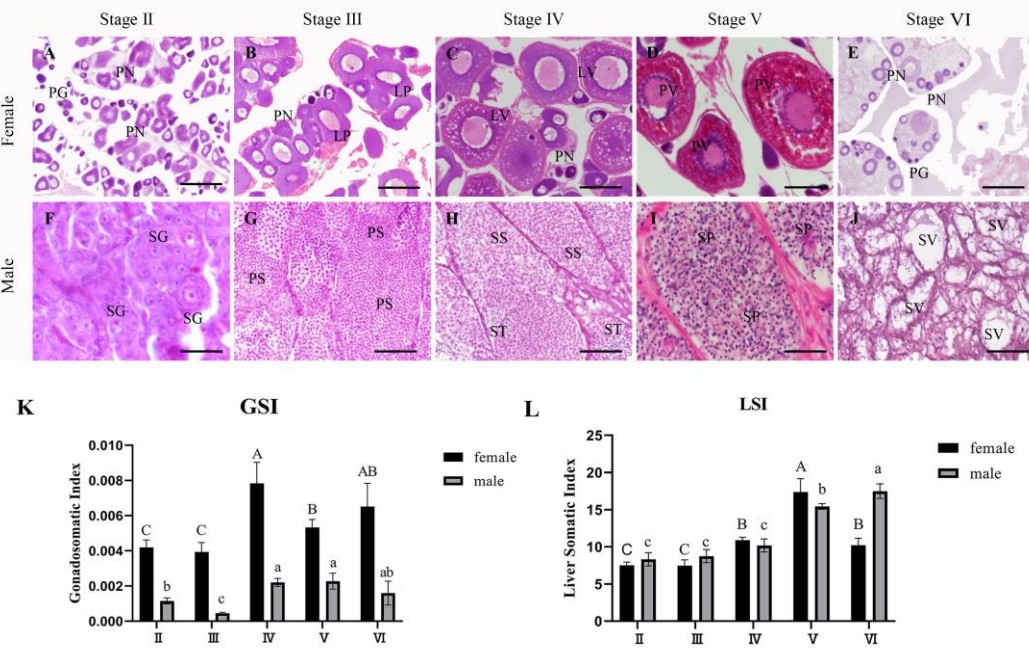

**Figure 1.** (**A–E**) Gonadal morphology of different developmental stages of turbot in females. (**F–J**) Gonadal morphology of different developmental stages of turbot in males. (**K**) GSI of the turbot during different gonadal developmental stages. (**L**) LSI of the turbot during different gonadal developmental stages. LP, late perinucleolus; LV, late vitellogenic oocyte; PG, primary growth oocyte; PN previtellogenic oocyte; PS, primary spermatocyte; PV, post-vitellogenic oocyte. SG, spermatogonium; SP, spermatozoon; SS, secondary spermatocyte; ST, spermatid; SV, seminal vesicle. Scale bar, (**A–E,G,H,J**), 100 μm; (**F,I**), 100 μm. The results are expressed as the mean $\pm$ SEM. The different uppercase letters represent statistical significance ($p < 0.05$) between two stages in females. The different lowercase letters represent statistical significance ($p < 0.05$) between two stages in males.



The GSI and LSI were also calculated (Figure 1K,L). In female turbot, the GSI increased from stage II to IV and peaked at stage IV. Then, the GSI decreased at stage V when oocytes ovulated, followed by an increase at stage VI. The LSI showed a trend of increasing at first and then decreasing with the peak at stage V. In male turbot, the GSI peaked at stage V, and the LSI increased from stage II to stage VI.

### 3.2. Molecular Cloning of Turbot igf3

The cloned *igf3* cDNA is 2261 bp in full length, including an open reading frame (ORF) of 891 bp and a 3′ untranslated terminal region (UTR) of 1370 bp. The predicted protein contains 296 amino acids, including a signal peptide and 5 domains of B, C, A, D, and E, with 6 conserved cysteine residues in domains of B and A (Figure 2A). The turbot IGF1 (GenBank accession No. NC_049689.1) and IGF2 (GenBank accession No. NC_049689.1) were obtained from NCBI. An obvious IGF domain was involved in IGF1, IGF2, and IGF3 (Figures 2B and 3B) based on the sequence analysis. The modeled three-dimensional structure of the IGF1, IGF2, and IGF3 is visualized and shown in Figure 3C.

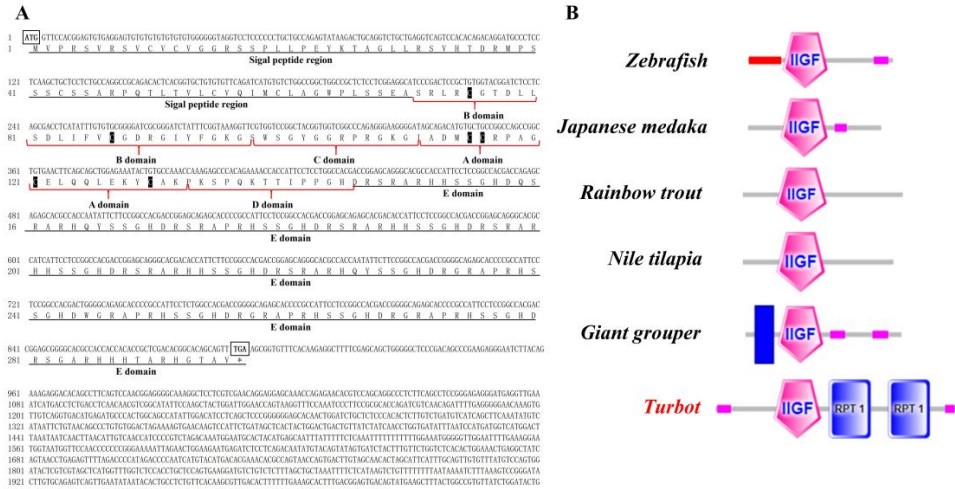

**Figure 2.** (**A**) Full sequence and structural domain of turbot *igf3*. The signal peptide and B, C, A, D, E domains are underlined. The six cysteine residue sites are marked with white letters on a black background. (**B**) Comparison of IGF3 domains of turbot and other teleost fish.

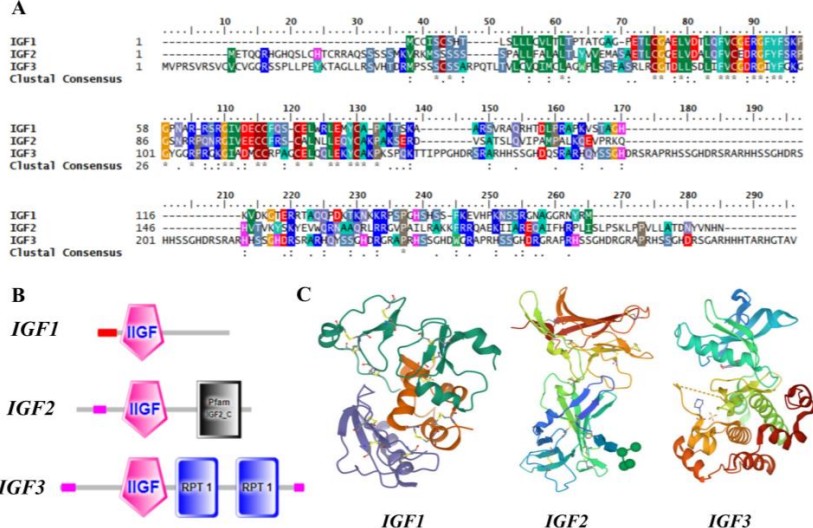

**Figure 3.** The amino acid alignments (**A**), structural domain (**B**), and three-dimensional structure (**C**) of IGF1, IGF2, and IGF3 of turbot.

### 3.3. Cross-Species Changes in igfs Members and Copy Numbers

A total of 68 *igf* genes were screened from NCBI in 23 representative teleost species (Table 2). Among these genes, the *igf2* typically consisted of two paralogs (*igf2a* and *igf2b*) in several fish, such as zebrafish and Atlantic salmon (*Salmo salar* L.). The *igf3* was usually found in Atherinomorpha, Protacanthopterygii, Ostariophysi, Clupeomorpha, Osteoglossomorpha, and Holostei.

**Table 2.** Identification of *igf* genes in teleost.

| Superorder | Species | Common Name | *igf1* | *igf2 (igf2a/igf2b)* Numbers | *igf3* |
|---|---|---|---|---|---|
| Percomorpha | *S.maximus* | Turbot | 1 | 1 | 1 |
| | *P.olivaceus* | Japanese flounder | 1 | 1 | - |
| | *C.semilaevis* | Half-smooth tongue sole | 1 | 1 | - |
| | *M.salmoides* | Largemouth bass | 1 | 1 | - |
| | *S.lucioperca* | Pikeperch | 1 | 2 | 1 |
| | *M.saxatilis* | Striped sea-bass | 1 | 1 | 1 |
| | *P.flavescens* | Yellow perch | 1 | 1 | 1 |
| | *E.lanceolatus* | Giant grouper | 1 | 1 | 1 |
| | *O.niloticus* | Nile tilapia | 1 | 1 | 1 |
| | *L.crocea* | Large yellow croaker | 1 | 1 | 1 |
| Atherinomorpha | *O. latipes* | Japanese medaka | 1 | 1 | 1 |
| | *X. maculatus* | Southern platyfish | 1 | 1 | 1 |
| Protacanthopterygii | *S. salar* | Atlantic salmon | 1 | 2 | 1 |
| | *O.mykiss* | Rainbow trout | 1 | 1 | 1 |
| Paracanthopterygii | *G. morhua* | Atlantic cod | 1 | 1 | - |
| Ostariophysi | *D. rerio* | Zebrafish | 1 | 2 | 1 |
| | *P.hypophthalmus* | Striped catfish | 1 | 2 | 1 |
| Clupeomorpha | *C. harengus* | Atlantic herring | 1 | 1 | 1 |
| Osteoglossomorpha | *S. formosus* | Asian arowana | 1 | - | 1 |

### 3.4. Phylogenetic and Synteny Analysis

Turbot IGF3 exhibited high identity to other teleosts, ranging from 75% to Japanese medaka, 74% to Zebrafish, 70% to channel catfish (*Ictalurus punctatus*), and 68% to rainbow trout (*Oncorhynchus mykiss*). Meanwhile, IGF1 exhibited 51.7% identity and 73.5% similarity with IGF2, 36.7% identity, and 69.8% similarity with IGF3. In addition, IGF2 showed 75.1% similarity with IGF3. The phylogenetic analysis revealed relatively higher degrees of identity of IGF2 and IGF3 among the analyzed fish species. While IGF1 showed a variety of clusters (Figure 4).

Synteny analysis of the *igf1*, *igf2*, and *igf3*, using turbot as the base reference, was performed to confirm their origin in fish and human species. Consistent with this, a total of 8 genes (*pmch*, *parpbp*, *nup37*, *th2*, *pah*, *itfg2*, *ascl1a*, and *nrip2*) neighbor the turbot *igf1* gene (Figure 5A); a total of 8 genes (*mrpl23*, *th*, *nap1|4a*, *phlda2*, *osbpl5*, *dusp8a*, *mob2a*) neighbor the turbot *igf2* gene (Figure 5B); a total of 6 gene (*stk38a*, *irif1*, *renbp*, *ndufb11*, *dedd* and *efhd2*) neighbor the turbot *igf3* gene (Figure 5C). It showed that the *igf1*, *igf2*, and *igf3* are relatively conserved in various fish species.

Humans lost larger numbers of genes next to *igf1*. Similarly, humans lost several genes next to *igf2*, which also happens in medaka. Moreover, the genetic relationship between turbot and zebrafish *igf1* is close. The genetic relationship between turbot and medaka *igf3* is close. The genetic relationship between turbot and zebrafish *igf2* is close.

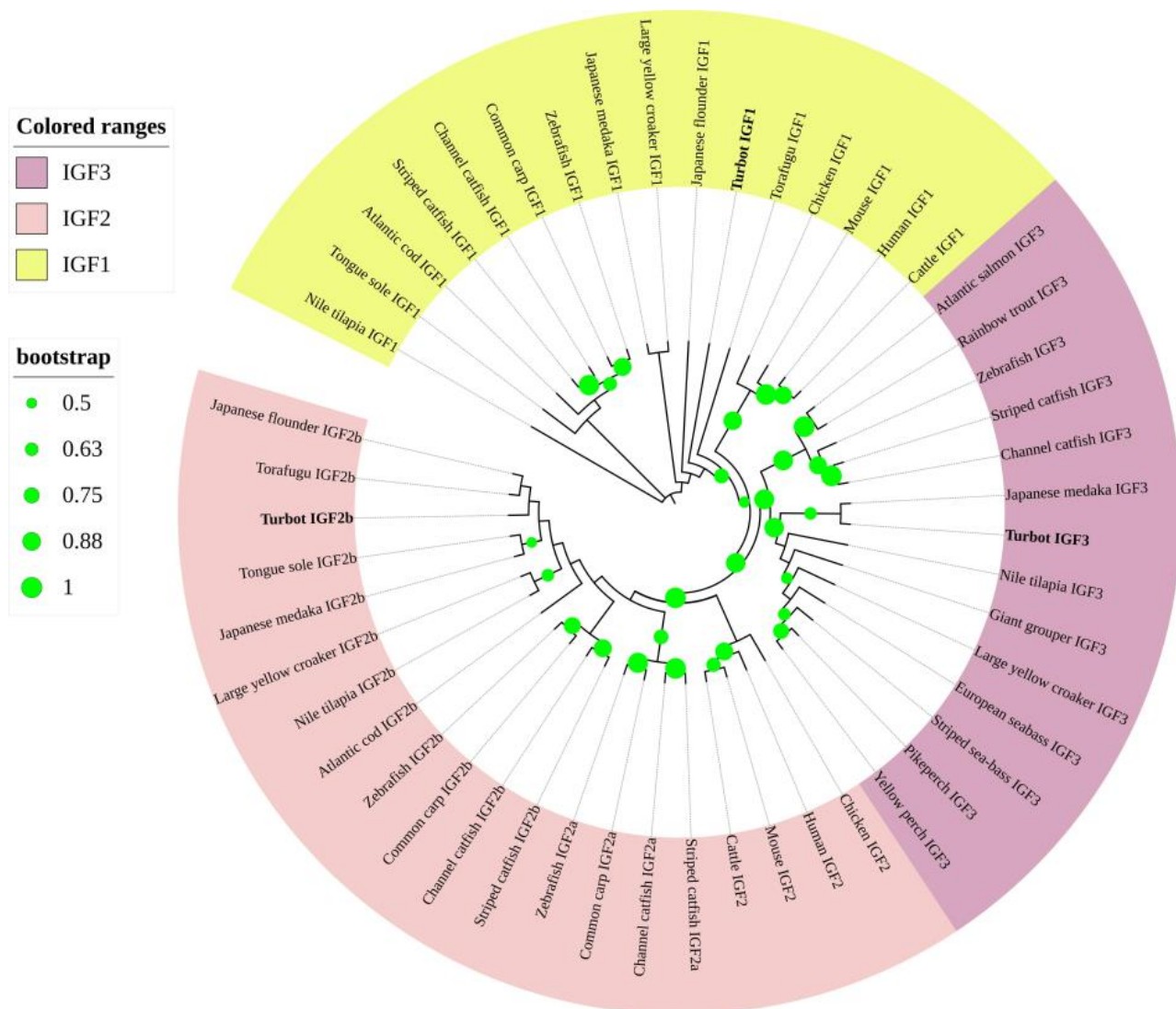

**Figure 4.** Phylogenetic analysis of IGFs from different teleosts. Turbot IGF1 (XP_035474357.1), Japanese flounder IGF1 (XP_019960503.1), Tongue sole IGF1 (NP_001281127.1), Channel catfish IGF1 (NP_001187224.1), Zebrafish IGF1 (NP_571900.1), Large yellow croaker IGF1 (NP_001290263.1), Atlantic cod IGF1 (XP_030209089.1), Torafugu IGF1 (NP_001177291.1), Chicken IGF1 (NP_001004384.1), Nile tilapia IGF1 (NP_001266432.1), Cattle IGF1 (NP_001071296.1), Japanese medaka IGF1 (XP_023808207.1), Human IGF1 (NP_000609.1), Mouse IGF1 (NP_001104744.1), Common carp IGF1 (XP_018948511.1), Striped catfish IGF1 (XP_034169273.1), Turbot IGF2 (XP_035498746.1), Japanese flounder IGF2 (XP_019957746.1), Half-smooth tongue sole IGF2 (NP_001281148.1), Channel catfish IGF2 (NP_001187875.1), Channel catfish IGF2 (NP_001187128.1), Zebrafish IGF2 (NP_571508.1), Zebrafish IGF2 (NP_001001815.1), Large yellow croaker IGF2 (XP_019124122.1), Atlantic cod IGF2 (XP_030221784.1), Torafugu IGF2 (XP_003967407.1), Chicken IGF2 (NP_001025513.1), Nile tilapia IGF2 (NP_001266572.1), Cattle IGF2 (NP_001354556.1), Japanese medaka IGF2 (XP_023811944.1), Human IGF2 (NP_000603.1), Mouse IGF2 (NP_001116208.1), Common carp IGF2 (XP_018955405.1), Striped catfish IGF2 (XP_026793492.1), Striped catfish IGF2 (XP_026775548.1), Turbot IGF3 (XP_035500204.1), Channel catfish IGF3 (XP_017323682.1), Zebrafish IGF3 (NP_001108522.1), Large yellow croaker IGF3 (XP_010749392.1), Rainbow trout IGF3 (XP_021471565.1), Giant grouper IGF3 (XP_033486016.1), Japanese medaka IGF3 (XP_011476003.1), Yellow perch IGF3 (XP_028439487.1), Striped sea-bass IGF3 (XP_035513051.1), Atlantic salmon IGF3 (XP_014001555.1), Pikeperch IGF3 (XP_031167391.1), Striped catfish IGF3 (XP_026784148.2).

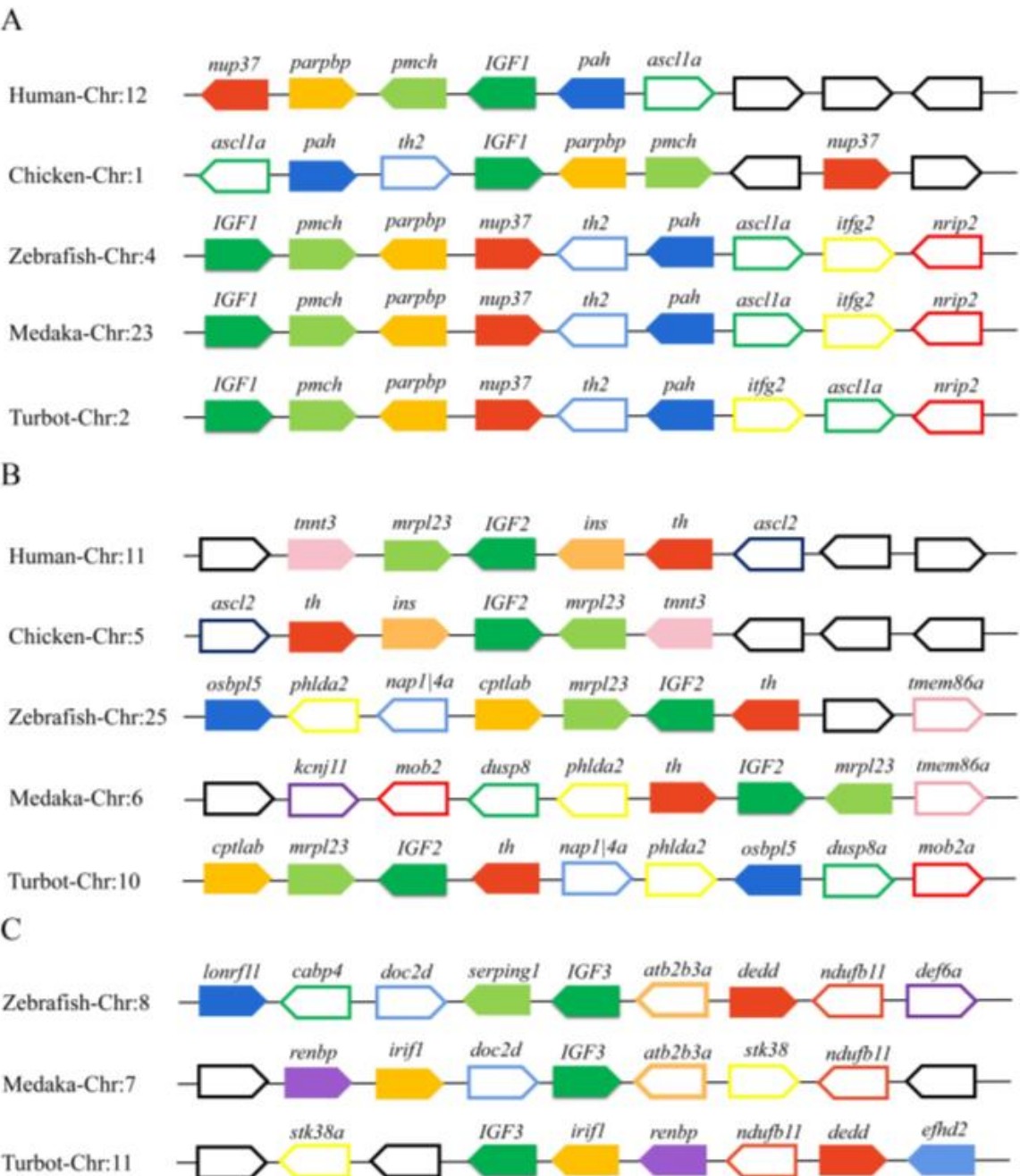

**Figure 5.** Syntenic analysis of *igf1* (**A**), *igf2* (**B**), *igf3* (**C**) in turbot and other species. The *igfs* are shown in green. The other genes are shown in different colors.

*3.5. Tissue Expression Pattern of igf3 mRNA*

All the tissues of the testis, ovary, liver, brain, heart, kidney, muscle, pituitary, eye, intestines, gill, skin, spleen, and stomach of male and female turbot were obtained. The expression levels of turbot *igf3* transcripts were detected by qPCR. As shown in Figure 6, the *igf3* transcript was detected in all analyzed tissues. The expression level of *igf3* in the gonad was significantly higher than in other tissues, followed by the liver, brain, heart, and pituitary gland. The expression pattern was similar in males and females.

The expression levels of *igf3* transcript in five brain structures (cerebellum, hypothalamus, pituitary gland, telencephalon, and diencephalon) of male turbot were also analyzed. As shown in Figure 7, *igf3* mRNA showed high expression levels in the hypothalamus and

pituitary, and the levels of *igf3* in the hypothalamus were the highest among all structures of the brain.

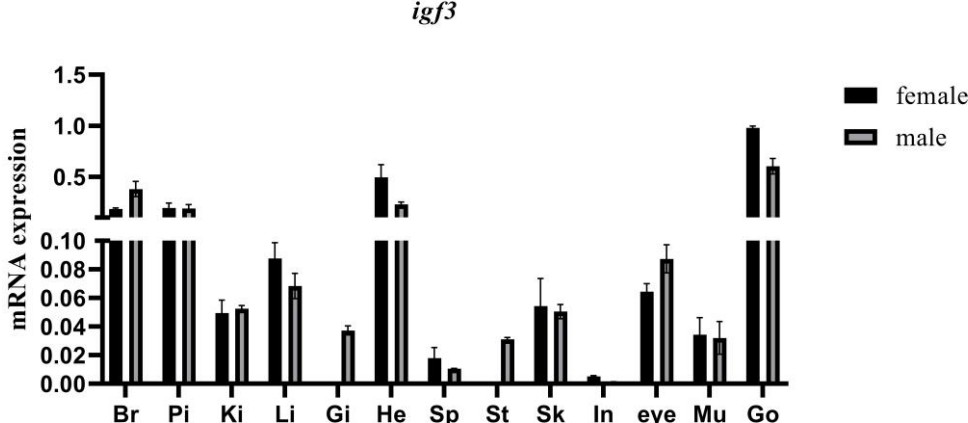

**Figure 6.** Tissue expression pattern of *igf3* in female and male turbot. Tissues abbreviation: brain (Br), pituitary (Pi), kidney (Ki), liver (Li), gill (Gi), heart (He), spleen (Sp), stomach (St), skin (Sk), intestine (In), eye(eye), muscle (Mu), gonad (Go). The results are expressed as the mean ± SEM.

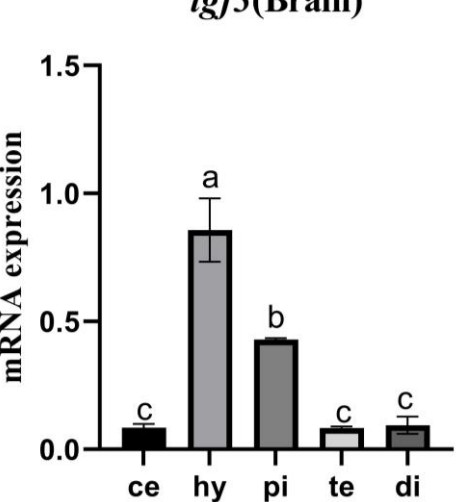

**Figure 7.** Expression pattern of *igf3* in the brain structures of male turbot. Abbreviation: cerebellum (ce), hypothalamus (hy), pituitary gland (pi), telencephalon (te) and diencephalon (di). The results are expressed as the mean ± SEM. Statistical significance between two different brain structures ($p < 0.05$) marked with different letters.

### 3.6. Expression Patterns igfs in Gonad and Liver during Gonadal Development

According to the previous research on gonadal morphology [2,16] and histology, the relative expression of *igfs* mRNA during gonadal development in the gonad and liver of both sexes of turbot was examined.

#### 3.6.1. Gonad

The expression levels of *igf1* mRNA gradually decreased from stage II to stage III, with the lowest values at stage III, and then significantly increased to stage VI, with the highest levels at stage VI in females and males of turbot (Figure 8A,B). The expression level of *igf2* mRNA remained unchanged from stage II to stage IV, and gradually increased from stage IV to stage VI in female turbot (Figure 8E). Additionally, *igf2* mRNA significantly increased from stage II to stage VI in male turbot (Figure 8F).

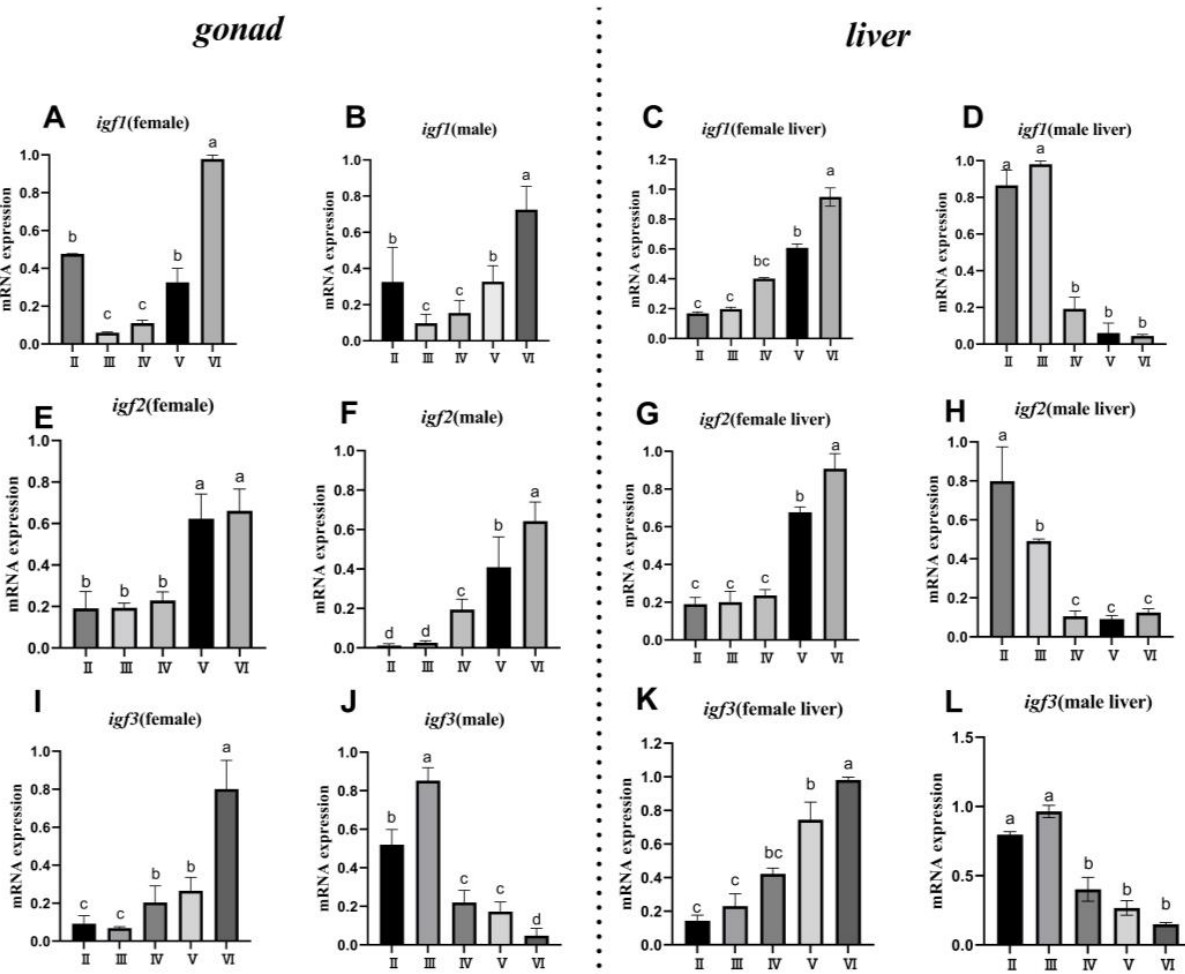

**Figure 8.** The expression of *igfs* mRNAs in gonad and liver during gonadal development of turbot. (**A**,**E**,**I**) The expression of *igfs* mRNAs in female gonad; (**B**,**F**,**J**) The expression of *igfs* mRNAs in male gonad; (**C**,**G**,**K**) The expression of *igfs* mRNAs in female liver; (**D**,**H**,**L**) The expression of *igfs* mRNAs in male liver. The results are expressed as the mean ± SEM. Statistical significance between two development stages (*p* < 0.05) marked with different letters.

For the expression of *igf3* in the ovary of turbot, it showed an increasing trend during gonadal development, with the lowest levels at stage III, and the highest levels at stage VI (Figure 8I). In testis, the highest level of *igf3* was at stage III (Figure 8J), followed by a decreasing trend from stage III to stage VI, with the lowest levels at stage VI.

### 3.6.2. Liver

The *igf1* mRNA levels in the liver gradually increased from stage II to stage VI, peaking at stage VI in female turbot (Figure 8C). In contrast, in male turbot, the *igf1* mRNA levels significantly increased from stage II to stage III and reached to the lowest value at stage VI (Figure 8D). The *igf2* mRNA remained unchanged from stage II to stage IV, but gradually increased from stage IV to stage VI, with the highest values observed at stage VI in female turbot (Figure 8G). In addition, for males, *igf2* mRNA significantly decreased from stage II to stage VI, and there was no significant difference from stage III to stage VI (Figure 8H).

As shown in Figure 8K, there is an increase in *igf3* expression in the female liver between stages II and VI, being the highest expression at stage VI. Additionally, in the male liver in Figure 8L, the expression of *igf3* mRNA increased from stage II to stage III, reaching the highest levels at stage III and dropping to the lowest level at stage VI.

## 4. Discussion

To date, the IGF system has been reported in more than 23 fish species, and three *igf* genes, *igf1*, *igf2* (*igf2a*, *igf2b*), and *igf3* have been identified in teleost. The *igf3* is a novel gonad-specific isoform that identified in several teleost species, including tilapia, zebrafish, common carp (*Cyprinus carpio*), and medaka [7,13]. In this study, we isolated a new member *igf3* in turbot. The predicted protein of turbot IGF3 has typical characteristics of the IGF system, which showed five conserved domains and six cysteine residues, similar to the IGF3 protein structure in common carp [17]. Turbot IGF3 has cysteine residues in the domains of B, C, A, D, and E, and amino acid residues for binding to IGF receptors [18]. Conversely, in orange-spotted grouper, it has not only five domains but also seven conserved cysteine residues [14]. While in zebrafish and tilapia, the IGF peptides consisted of only four domains [13] and six conserved cysteine residues [10]. These peptides shared high similarity with human IGF in their primary structure. All these results show that different fish species have taken different evolutionary branches, resulting in differences in specific protein structures. However, phylogenetic analysis showed that IGF3 exhibits a relatively low sequence homology among teleost, and the evolutionary diversity of IGF3 may be due to different genome duplications. It is interesting to note that IGF3 appears to have a closer relationship to IGF1 in the phylogenetic tree. This might be caused by whole genome duplication in vertebrates [19].

It was found that the predicted amino acid of IGF3 in turbot contained a transmembrane domain, similar to that of the silver pomfret, indicating that IGF3 of turbot is located in the cell membrane. While the proteins of IGF1 and IGF2 do not have transmembrane domains [8,16,20], which indicated that the protein of IGF1 and IGF2 are located in the cytoplasm. Generally, among all proteins found in eukaryotic organisms, one-third were transmembrane proteins [19]. Additionally, there were two types of transmembrane proteins, the bitopic proteins, and the polytopic proteins. The type of bitopic proteins crosses the membrane only once, while the type of polytopic proteins crosses the membrane multiple times. Each type of cell membrane reflected a special function [17]. In this study, the structural analysis demonstrated that the IGF3 of turbot belonged to the single-span membrane protein [19], which contained most of the growth factors and cell factors. These factors made them effective by binding to specific protein molecules to regulate cell-to-cell signaling, suggesting that IGF3 of turbot might also play a crucial role in intercellular signal transmission.

For sequence alignment and phylogenetic analysis, both IGF1 and IGF2 have similar protein sequences and domains [8]. In addition, their structures are also similar in zebrafish [9,21] and sliver pomfret [19]. This indicates that different amino acid sequences can form similar binding sites after protein folding, which can explain the similar functions of IGF1 and IGF2. It was also found that the IGF1 and IGF2 of turbot shared the highest homology with Japanese flounder and tongue sole, with a homology of 92% and 89%, respectively. This is in accordance with previous studies [2,8,15,20].

In this study, the transcript levels of *igf3* were high in the testis, ovary, brain, and pituitary. Taken with other studies in fish, such as common carp [17], zebrafish [10], and medaka [18], it is suggested that *igf3* might play a significant role in the gonadal development of teleost. While in silver pomfret (*Pampus argenteus*), the transcripts of *igf3* were specifically detected in the testis and ovary, suggesting that the expression of *igf3* is limited to fish species and is gonad-specific. Additionally, the *igf3* mRNA was also detected in the brain, especially in the hypothalamus and pituitary gland, which is consistent with that in tilapia [12], orange-spotted grouper [14], and silver pomfret [19]. Therefore, it could be confirmed that the *igf3* mRNA is expressed in the tissues of the BPG axis, indicating that it might play an important role in gonadal development involving the BPG axis.

In most teleost, the liver has been considered the main organ for synthesizing and targeting *igfs* [7]. At the same time, gonadal development is closely related to the liver [2,22]. Studies have demonstrated that paracrine *igfs* mediated the specific ovarian and spermatic development in several fish species [7,11,12,19]. The highest levels of *igf3* mRNA expression

were observed during oocyte maturation and yolk formation in zebrafish [11,16] and common carp [17]. Moreover, the *igf3*, in adult zebrafish testis, played roles in stimulating the proliferation of spermatogonia [23] and had a positive effect on reproduction in male tilapia by 17a-ethinylestradiol (EE2) [12]. What is more, it is also observed that *igf3* is involved in gonadal development, especially at the onset of meiosis in tilapia [12]. However, in common carp, the levels of *igf3* mRNA peaked during the testis recession period [17], indicating that it also affects sperm release. All the results suggest that the regulation of gonadal *igf3* may differ among species [24,25]. This study showed that the expression of *igf3* mRNA exhibit the highest level at stage VI in females, which supports a potential role of *igf3* in oocyte maturation and degradation [16,17]. In males, *igf3* mRNA maintained significantly high expression levels at stages II and III when the spermatogonia began to proliferate. So, it is speculated that the *igf3* might play a role in differentiating spermatogonia [26]. Yet, further experimental work to investigate should be required.

There are two other members in the IGF system, *igf1* and *igf2*, which have also been proven to express in gonads of many teleosts, and the regulation and function of IGFs in fish gonads have been summarized in Table 3. Generally, the expression levels of *igf1* and *igf2* vary depending on the developmental stage [27]. In turbot, the levels of *igf1* and *igf2* showed an increase during oogenesis and spermatogenesis, leading to the maturation and release of oocytes and sperm. Meanwhile, in the livers of both sexes, the expression levels of *igf1* and *igf2* are similar to that in the gonads. In most vertebrates, the liver has been confirmed as the main synthesis and target organ for IGFs [2,19], and the hypothesis that IGFs directly participate in the synthesis of nutrient substances for gametogenesis formed [26]. However, in turbot, there is a difference in male and female lives. In male livers, the highest levels of *igfs* happened at stages II and III, when spermatogenesis was beginning. While in female livers, it showed an increasing trend with the highest level in stage VI. So, it is speculated that more nutrient substances are needed during oocyte maturation in female and spermatogonia differentiation. What is more, studies have demonstrated that the expression levels of *igf2* mRNA were always higher than those of *igf1* in the gonads and livers of both sexes in some fishes, such as silver pomfret [19] and rainbow trout [28], suggesting that *igf2* possess more important role than *igf1* in gametogenesis. Further research on the detection and location of *igf1* and *igf2* is necessary to determine their specific role in gonadal development. In addition, the expression of *igf1* and *igf2* mRNA of turbot was also detected during the unfertilized egg stage and post-larva stage, indicating that *igfs* may have a function in cell growth and division in turbot [8]. During the metamorphosis of turbot, *igf1*, rather than *igf2*, may play a crucial role in regulating the metamorphic development, as the mRNA *igf1* showed significantly higher levels at the early metamorphosis stage, followed by a decrease until the metamorphosis was completed [20]. Therefore, as an important regulator of growth and development, *igf1* and *igf2* are involved in many regulatory processes in turbot.

**Table 3.** Studies on IGF system in teleost.

| Genes | Number of Species | Function | References |
|-------|------------------|----------|------------|
| *igf1* | 16 | Ovary: follicular cell proliferation follicle growth, oocyte maturation. Testis: stimulation of spermatogenesis. | *C. semilaevis* [29]; *D. rerio* [30]; *E.lanceolatus* [31]; *G.aculeatus* [32]; *L.crocea* [33]; *M.salmoides* [34]; *O.latipes* [35]; *O.mykiss* [28]; *O.niloticus* [36]; *P.flavescens* [37]; *P.hypophthalmus* [38]; *P.olivaceus* [39]; *S.lucioperca* [40]; *S.maximus* [41]; *S. salar* [42]; *X. maculatus* [43]; |

**Table 3.** *Cont.*

| Genes | Number of Species | Function | References |
|---|---|---|---|
| *igf2* (*igf2a/igf2b*) | 11 | Ovary: oocyte maturational competence, oocyte maturation. Testis: DNA synthesis of spermatogonia and spermatocytes. | *C. semilaevis* [29]; *D. rerio* [30]; *E.lanceolatus* [21]; *G.aculeatus* [32]; *L.crocea* [33]; *M.salmoides* [34]; *O.mykiss* [28]; *O.niloticus* [36]; *P.flavescens* [44]; *P.hypophthalmus* [38]; *S.maximus* [15]; *X. maculatus* [43]. |
| *igf3* | 9 | Ovary: follicular cell survival, follicular cell development and ovulation. Testis: proliferation and division of type A spermatogonia and proliferation of Sertoli cells during spermatogenesis. | *D. rerio* [23]; *E.lanceolatus* [45]; *M.salmoides* [34]; *O.latipes* [18]; *O.mykiss* [45]; *O.niloticus* [46]; *P.flavescens* [44]; *P.hypophthalmus* [38]; *S.salar* [47]. |

## 5. Conclusions

In conclusion, we have successfully cloned a new specific *igf3* in turbot, and it showed a highly similar structure to *igf3* to other teleosts. The high expression levels of *igf3* happened in the ovaries and testis, which strongly suggests that it might play an important role in oogenesis and spermatogonial development. Additionally, along with *igf1* and *igf2*, all three *igfs*, in the liver and gonad, are likely to be involved in regulating the gonadal development in both male and female turbot. These results can improve our understanding of the IGF system in the regulation of gonadal development in turbot and provide additional information on IGFs in teleost.

**Author Contributions:** Conceptualization, C.Z. and Y.R.; methodology, S.Z.; investigation, S.Z.; resources, Y.R.; data curation, S.Z., M.W., and Y.D.; writing—original draft preparation, S.Z.; writing—review and editing, C.Z. and Y.R.; visualization, S.Z.; supervision, C.Z. and Y.R.; project administration, Y.R.; funding acquisition, C.Z. and Y.R. All authors have read and agreed to the published version of the manuscript.

**Funding:** This research was funded by the Natural Science Foundation of China (No. 31802319), the Key Research and Development Program of Shandong Province (No. 2021LZC027), Advanced Talents Foundation of QAU (No. 6631119055), and it was also supported by "First class fishery discipline" program in Shandong Province, China.

**Institutional Review Board Statement:** All experimental procedures were conducted strictly in accordance with the research protocols approved by Qingdao Agricultural University for the ethical treatment of experimental animals.

**Informed Consent Statement:** Not applicable.

**Data Availability Statement:** Data will be made available on request.

**Conflicts of Interest:** The authors declare no conflict of interest.

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
