# Peer review of "Identification of a New Insulin-like Growth Factor 3 (igf3) in Turbot (Scophthalmus maximus): Comparison and Expression Analysis of IGF System Genes during Gonadal Development"

_fishes, doi:10.3390/fishes8050240_

Round 1

Reviewer 1 Report

Dear authors,
I have read your article describing the structure and expression of igf3 in turbot. This is an extremely interesting and novel topic due to the limited known mechanism of action of the gene. After reviewing the paper, I believe that it needs some corrections and clarifications:
For explanation:
Why was sequencing carried out on male gonads and not both sexes? Or in the liver?
The paper is also missing the most important part of the discussion that needs to be added - nt the expression of this gene in the brain and how the data obtained relate to the literature data.
To be improved:
Furthermore, I draw your attention to numerous stylistic and punctuation errors (typos, missing capital letters at the beginning of sentences, etc.). In some passages, the vocabulary used is not specialised.
In some cases, there has also been erroneous citation - e.g. of publications that do not mention the aspect cited in the text. Please verify citations and literature data.
Recent publications are 2017 upwards. papers from 2008-2010 cannot be called "recent"

Sincerely,

R2.

Author Response

Dear reviewer,

Thanks very much for your consideration and careful work for our manuscript (previous manuscript number: fishes-2340287). The suggestions are very valuable. We have revised the manuscript according to the reviewer’s comments. We really like it to be published in “fishes”.

Following are our revision statements on the manuscript amending.

To be improved:

  1. Why was sequencing carried out on male gonads and not both sexes? Or in the liver?

Response: We would apologize for the confusion here. Actually, the expression of igf3 is predominantly confined to the gonads of male and female fish, including zebrafish, tilapia, medaka, and silver pomfret, with some weak expression in other tissues such as brain. For turbot, we cloned the igf3 gene using male gonads, and obtained the full-length of igf3 using 5’ and 3’ RACE. Once obtained the full length, we designed primer of igf3 to exam it in female gonads and liver, and confirmed that the cloned fragment was consistent with the obtained clone in male gonad, indicating that it is the igf3 gene of turbot. So, in the manuscript of method, the male gonad was used for the sequencing.

  1. The paper is also missing the most important part of the discussion that needs to be added - the expression of this gene in the brain and how the data obtained relate to the literature data.

Response: We are so sorry about it. Based on the tissue expression pattern, we found that the igf3 mRNA is highly expressed in the gonad, brain, and liver. And in previous studies of tilapia, common carp, and orange-spotted grouper, the igf3 transcripts were detected in areas of brain, especially the telencephalon and mesocerebrum. Therefore, it could be confirmed that the igf3 mRNA is expressed in the tissues of the hypothalamic-pituitary-gonadal axis, which suggests that it might play an important role in gonadal development involving of the hypothalamic-pituitary-gonadal axis. Further, if it is possible, more experiments could be done to confirm the expression and the function in the brain in these fishes. The information about the expression in brain was supplemented in the discussion section in Line 336-341, “Additionally, the igf3 mRNA was also detected in the brain, especially in the hypothalamus and pituitary gland, which is consistent with that in tilapia [12], orange-spotted grouper [14] and silver pomfret [19]. Therefore, it could be confirmed that the igf3 mRNA is expressed in the tissues of the BPG axis, indicating that it might play an important role in gonadal development involving of the BPG axis.”

To be improved:

  1. Furthermore, I draw your attention to numerous stylistic and punctuation errors (typos, missing capital letters at the beginning of sentences, etc.). In some passages, the vocabulary used is not specialised.

Response: We are sorry about the shortcomings. The whole manuscript has been checked carefully, and all the mistakes have been revised. The vocabulary used also replaced by academic vocabulary. What’s more, our manuscript has been polished by an English native speaker.

  1. In some cases, there has also been erroneous citation - e.g. of publications that do not mention the aspect cited in the text. Please verify citations and literature data.

Recent publications are 2017 upwards. papers from 2008-2010 cannot be called "recent"

Response: We are sorry about the erroneous in citation. All the references have been checked, and the errors in the citation have been revised. Meanwhile, we checked the year of publication that cited in our manuscript, and the incorrect description about “recent”, “recently” and “previous” have all been revised. In addition, two references have been replaced to the latest references in Line482-487. The references are list following:

  1. Yan, R., Ding, J., Yang, Q., Zhang, X., Han, J., Jin, T., An, Y. Lead acetate induces cartilage defects and bone loss in zebrafish embryos by disrupting the GH/IGF-1 axis. Ecotoxicol Environ Saf, 2023, 253, 114666. https://doi.org/10.1016/j.ecoenv.2023.114666
  2. Hanson, A. M., Kittilson, J. D., & Sheridan, M. A. Environmental estrogens inhibit insulin-like growth factor (IGF) receptor mRNA expression, IGF binding, and IGF signaling ex vivo in rainbow trout (Oncorhynchus mykiss). Gen Comp Endocrinol. 2023, 330, 114125. https://doi.org/10.1016/j.ygcen.2022.114125

Reviewer 2 Report

The authors describe the expression of a novel igf in turbot. Authors make genetic comparisons and check the expression levels of igf3 at different stages of gonadal development. This is a descriptive story but very valuable. Results are nicely shown. My most important comment is related to the first figure where I think authors should explain better how they did the histology.

I have some other comments:

Abstract

-       Line 21. Please check sentence “…during all through process..”

Introduction

-       The liver has been reported to regulate fish gonad development by modulating the BPG axis. Please include references

Results

-       The histological characterization of the gonads is quite nice. However, the staining protocol is missing. I am assuming this is H&E but how the sections were obtained and stained should be included in the methods. Also, How the authors distinguish between the different stages of the gametes? Differences can be observed, more or less, in case of oocyte development but I cannot see in the manuscript how the authors distinguish between ST, SP, SS… Since establishing the differences between stages is key for their story, it would be nice if authors could explain this figure further

-       Also, in figure 1K and L, letters mark statistical significance, but between what? Authors need to clarify the meaning of each letter. The same happens in Figure 7

-       In figure 5C it would be nice to also have the comparison of the gene location in human as the authors show for IGF1 and IGF2. I find a little bit confusing that the authors choose different species in each group. I would say that human, chicken, zebrafish and medaka are a must for all igfs, unless there is a reason.

-       I am not sure what the authors mean with “Tissue and brain internal structure distribution of igf3 mRNA”. It reads weird. Authors are checking the relative expression of igf3 in different organs. Distribution of mRNA is typically addressed using in situ hybridization. Title should indicate this.

-       Please, check sentence in line 224-225

-       In Figure 7 and 8, the legend is explained both in the X axis and in the right part of the graph. This repetition has no point. It is ok in figure 7 but figure 8 is really small. I would suggest removing the label on the right part of each graph and make them a little bit bigger. Once again, please clarify the meaning of the letters. Stating “Statistical significance (p<0.05) marked with different letters” is not enough

-       In the description of figure 8 the authors say: “As shown in Figure 8K, the expression of igf3 mRNA in liver of female increased from stage II to stage III, reaching the highest levels at stage III and dropping to the lowest level at stage VI. And the igf3 mRNA in male liver had higher levels stage II and III than stage IV to VI (Figure 8L)”. However, there is an increase in ifg3 expression in the female liver between stages III and VI, being the highest expression at stage VI. What is it then?

Discussion

-       Line 288, I think you mean protein structure, and not gene structure, since you are talking about peptides. If you mean gene, it would be necessary to amend the nomenclature of the paragraph

-       Line 292, what do you mean by whole genome replication? Do you mean duplication?

-       Please, be careful with sentences such as: “In males, igf3 maintained significant high expression levels during the proliferation of spermatogonia at stages II and III, resulting in increased production of differentiating spermatogonia and ultimately stimulation of spermatogenesis” Correlation doesn´t mean causation. There is no evidence supporting this sentence. I would agree that the coincidence of the high expression of igf3 and the stage would suggest that, but unless you have evidence from mutants or knockdowns, I would recommend to tone down these sentences. There are several examples in the discussion.

-       I think the discussion is a little bit too long due to all the repetition of the results, but this is just an opinion. However, I am a little surprised that the authors do not discuss much about the huge sex-related differences in the expression of all three igfs in the liver. In my opinion this is a very interesting result that could help understanding the role of igfs in different males and females

Author Response

Dear reviewer,

Thanks very much for your consideration and careful work for our manuscript (previous manuscript number: fishes-2340287). The suggestions are very valuable. We have revised the manuscript according to the reviewer’s comments. We really like it to be published in “fishes”.

Following are our revision statements on the manuscript amending.

  1. Abstract

-Line 21. Please check sentence"…during all through process.

Response: We are so sorry about the mistake statement here. The meaning of this sentence focused on the expression during all the process of oogenesis. So, the sentence has been revised in Line 21, “In the gonad and liver of female turbots, the expression levels of igfs mRNA significantly increased from stage II to VI during the process of oogenesis, including maturation and degeneration.”

  1. Introduction

- The liver has been reported to regulate fish gonad development by modulating the BPG axis. Please include references.

Response: Thanks for your suggestion. Generally, the gonadal development in teleost is related to the brain and liver. On the one hand, in the female’s liver, some necessary substance, such as vitellogenin, are typically synthesized and taken up by developing oocytes. In addition, more research in BPG-liver axis in fish reproduction and ovarian maturation have been investigated. In C. semilaevis ovary maturation, it has been confirmed that miR-186-x directly regulates igf2r expression, modulates the reproductive endocrine system. Therefore, it would be a better understanding the transcriptomic modulation of the BPG axis is related liver to affect gonadal development. The references have been added in the manuscript in Line 43.

  1. Xue, X. Wang, S. Xu, Y. Liu, C. Feng, C. Zhao, Q. Liu, and J. Li, Expression profile and localization of vitellogenin mRNA and protein during ovarian development in turbot (Scophthalmus maximus), Comp Biochem Physiol B Biochem Mol Biol. 2018, 226, 53-63. https://doi.org/10.1016/j.cbpb.2018.08.002
  2. Kwintkiewicz, and L. C. Giudice, The interplay of insulin-like growth factors, gonadotropins, and endocrine disruptors in ovarian follicular development and function, Semin Reprod Med. 2009,27, 43-51. https://doi.org/10.1055/s-0028-1108009
  3. Zhang, B. Shi, P. Shao, C. Shao, C. Wang, J. Li, X. Liu, X. Ma, X. Zhao, The identification of miRNAs that regulate ovarian maturation in Cynoglossus semilaevis. Aquaculture. 2022, 555, 738250. https://doi.org/10.1016/j.aquaculture.2022.738250
  4. Results

-       The histological characterization of the gonads is quite nice. However, the staining protocol is missing. I am assuming this is H&E but how the sections were obtained and stained should be included in the methods. Also, How the authors distinguish between the different stages of the gametes? Differences can be observed, more or less, in case of oocyte development but I cannot see in the manuscript how the authors distinguish between ST, SP, SS… Since establishing the differences between stages is key for their story, it would be nice if authors could explain this figure further

Response: Thanks very much for the suggestion. Firstly, we supplemented the method in the section of 2.2 Histological observation in Line 87-91, “The fixed gonadal tissues were dehydrated by a series of gradient alcohol, transparented by xylene, and embedded by paraffin. Then the paraffin blocks were cut into 5mm sections. After stained with hematoxylin and eosin, the slices were observed with a microscope (Axioscope 5, Zeiss).”

In addition, we are so sorry about the shortcoming in Figure 1. the gonadal development stages were distinguished by the morphology and characteristics of germ cell that referred in previous studies. We have supplemented the statement in results of 3.1 in Line 134-149.

-       Also, in figure 1K and L, letters mark statistical significance, but between what? Authors need to clarify the meaning of each letter. The same happens in Figure 7

Response: Thanks very much for the suggestion. In Figure 1, The different uppercase letters represent statistical significance (p < 0.05) between two stages in female. The different lowercase letters represent statistical significance (p < 0.05) between two stages in male. And in Figure 7, Statistical significance between two different brain structures (p<0.05) marked with different letters. The statement has been revised in the figure caption.

  1. Results

-       In figure 5C it would be nice to also have the comparison of the gene location in human as the authors show for IGF1 and IGF2. I find a little bit confusing that the authors choose different species in each group. I would say that human, chicken, zebrafish and medaka are a must for all igfs, unless there is a reason.

Response: Thank you very much for the suggestion. we have checked the figure, and chosen the same species in igf1, igf2, igf3.  The human, chicken, zebrafish and medaka were included. However, for igf3, it is a type that only found in fish. So, in the Figure 5C, the human and chicken were not present. We have revised the Figure 5 in the manuscript in Page 8.  

-       I am not sure what the authors mean with “Tissue and brain internal structure distribution of igf3 mRNA”. It reads weird. Authors are checking the relative expression of igf3 in different organs. Distribution of mRNA is typically addressed using in situ hybridization. Title should indicate this.

Response: Thanks very much for the suggestion. The result of 3.5 is about the expression of igf3 mRNA in different tissues and different structures of brain. The statement has been revised in Line 236 and the Results of 3.5.

  1. -       Please, check sentence in line 224-225

Response: We are so sorry about it. The sentence has been checked carefully and revised in Line 240-241.

  1. -       In Figure 7 and 8, the legend is explained both in the X axis and in the right part of the graph. This repetition has no point. It is ok in figure 7 but figure 8 is really small. I would suggest removing the label on the right part of each graph and make them a little bit bigger. Once again, please clarify the meaning of the letters. Stating “Statistical significance (p<0.05) marked with different letters” is not enough

Response: Thank you very much for the suggestion. in Figure 8, the right part was already removed. And the statement in figure caption has been revised in Line 274-276.

  1. -       In the description of figure 8 the authors say: “As shown in Figure 8K, the expression of igf3 mRNA in liver of female increased from stage II to stage III, reaching the highest levels at stage III and dropping to the lowest level at stage VI. And the igf3 mRNA in male liver had higher levels stage II and III than stage IV to VI (Figure 8L)”. However, there is an increase in igf3 expression in the female liver between stages III and VI, being the highest expression at stage VI. What is it then?

Response: We are so sorry about the confusing here. The statements have been revised in the manuscript in Line 286-289, “As shown in Figure 8K, there is an increase in igf3 expression in the female liver between stages II and VI, being the highest expression at stage VI. And in male liver in Figure 8L, the expression of igf3 mRNA increased from stage II to stage III, reaching the highest levels at stage III and dropping to the lowest level at stage VI. As shown in Figure 8K, there is an increase in igf3 expression in the female liver between stages II and VI, being the highest expression at stage VI. And in male liver in Figure 8L, the expression of igf3 mRNA increased from stage II to stage III, reaching the highest levels at stage III and dropping to the lowest level at stage VI.”

  1. Discussion

-       Line 288, I think you mean protein structure, and not gene structure, since you are talking about peptides. If you mean gene, it would be necessary to amend the nomenclature of the paragraph

Response: Thanks very much for the suggestion. In this paragraph, we talk about the peptides, not the gene, and the statement has been revised in Line 303-304, “All these results show that different fish species have taken different evolutionary branches, resulting in differences in specific protein structure.”

  1. -       Line 292, what do you mean by whole genome replication? Do you mean duplication?

Response: It should be “duplication”, and we have revised in the manuscript in Line 308.

  1. -       Please, be careful with sentences such as: “In males, igf3 maintained significant high expression levels during the proliferation of spermatogonia at stages II and III, resulting in increased production of differentiating spermatogonia and ultimately stimulation of spermatogenesis” Correlation doesn´t mean causation. There is no evidence supporting this sentence. I would agree that the coincidence of the high expression of igf3 and the stage would suggest that, but unless you have evidence from mutants or knockdowns, I would recommend to tone down these sentences. There are several examples in the discussion.

Response: Thank you very much for the suggestion. The statement has been revised in the manuscript in Line 356-359, “In males, igf3 maintained significant high expression levels at stages II and III when the spermatogonia began to proliferation. So, it is speculated that the igf3 may be play role in increasing of differentiating spermatogonia. And it should be required further experimental to investigate. ”

In addition, we have checked the manuscript carefully, and the statements about the related description were all revised.

  1. -       I think the discussion is a little bit too long due to all the repetition of the results, but this is just an opinion. However, I am a little surprised that the authors do not discuss much about the huge sex-related differences in the expression of all three igfsin the liver. In my opinion this is a very interesting result that could help understanding the role of igfs in different males and females

Response: Thank you very much for the suggestion. The discussion has been revised. And the information about the sex-related differences in the expression of igfs in the liver has been supplemented in the manuscript in Line 366-373.

Round 2

Reviewer 1 Report

Dear Authors,

Thank you very much for all your work. The corrections and attached explanations are satisfying.

Sincerely,

R